# Global Inequities in COVID-19 Vaccination: Associated Factors and Tools to Measure Inequality

**DOI:** 10.3390/vaccines11071245

**Published:** 2023-07-16

**Authors:** Ubaldo Miranda-Soberón, Isabel Pino-Arana, Jeny del Rio-Mendoza, Mario Chauca

**Affiliations:** 1Human Medicine Faculty, National University “San Luis Gonzaga”, Ica 11004, Peru; ubaldo.miranda@unica.edu.pe (U.M.-S.); jeny.delrio@unica.edu.pe (J.d.R.-M.); 2Nursing Faculty, National University “San Luis Gonzaga”, Ica 11004, Peru; dpino@unica.edu.pe; 3Industrial Engineering, Engineering Faculty, Ricardo Palma University, Lima 15039, Peru

**Keywords:** vaccination, COVID-19, inequities, socioeconomic factors

## Abstract

Introduction: Socioeconomic factors have been recognized by the WHO as determinants of health, and it is important to consider these factors in decision making to curb existing inequality in vaccination for SARS-CoV-2, which causes COVID-19. Objective: We aimed to determine whether there is a correlation between socioeconomic factors and vaccination worldwide and measure inequality. Method: A study of secondary sources was carried out to assess inequality in vaccination against COVID-19 worldwide and its association with socioeconomic factors. For this assessment, 169 countries were chosen from January 2020 to March 2022 using LibreOffice and JASP 0.16.1.10. Several mathematical models and statistical tests were used, including a normality test, an analysis of frequencies and proportions, a Kruskal–Wallis test, Spearman’s correlations, a Lorenz curve, a Concentration Index, and a slope. Results: Correlations were found between socioeconomic factors and vaccination with one, two, and three doses. As the GDP showed correlations of 0.71 for one dose and 0.82 for three doses, we found that the greater the competitiveness of the countries, the higher the percentage of vaccinated individuals in their populations. According to the Concentration Index, there was greater inequality in vaccination with regard to receiving a higher number of doses, as reflected in the life expectancy indices of 0.16–0.19 and 0.50. The continent with the highest degree of inequality was Africa, and the continent with the lowest degree was America. South Americans were vaccinated with two doses at a rate of 6.19%/month, which was 4.3 times faster than Africans, with 72% of the population being vaccinated in South America, compared to only 16% in Africa. Conclusion: There is inequality in vaccination against COVID-19 with one, two, and three doses, which is associated with socioeconomic factors.

## 1. Introduction

The outbreak of SARS-CoV-2 began on 29 December 2019 in Wuhan, China; it is the causative agent of the disease defined by the World Health Organization as COVID-19, and had spread throughout the world [1]. It is characterized by genetic changes that produce new variants; these genetic changes induce permanent changes in the virus’s DNA sequences and can affect several nucleotides [2], resulting in an increase in its spread, greater virulence, and resistance to vaccines. Because of this, the US CDC [3] has classified its variants as variants under monitoring (VBM), variants of interest (VOI), variants of concern (VOC), and variants of high consequence (VOHC). At the end of 2020, the variants Delta (in the United Kingdom and South Africa, 2020) and Omicron (Brazil, 2022) [4] appeared. The emerging variants Beta, Delta, Lambda, and, more recently, Omicron, together with people’s refusal to receive the vaccine, caused an alarming increase in COVID-19 cases at the beginning of 2022 [5]. COVID-19 results in personal and social consequences and leads to health complications, with the most feared complications being pulmonary, extrapulmonary, and psychiatric symptoms, some of which may persist on a long-term basis [6,7,8]. Due to vaccination, the incidence of COVID-19 is expected to decrease along with mortality, as stated by Watson [9], who found a significant decrease in mortality between December 2020 and 2021. It is important to understand that vaccination is one of the most effective measures that can be taken against COVID-19, in addition to personal protection and hand washing. On the other hand, COVID-19 has pushed millions of people into extreme poverty, aggravating the impact of vulnerability on conflicts and violence and diminishing progress in almost all aspects of development. The effect of the pandemic has not been the same for all countries, with the impact being greater among the most vulnerable populations, who have suffered increased inequality, unemployment, and economic crises. It is crucial that countries have a recovery plan that involves the social economy and prioritize the most disadvantaged sectors [10].

Since vaccination with the Pfizer–BioNTech vaccine began in December 2020, 11 different vaccines have been administered worldwide, with others being under investigation [11]. Vaccines protect against COVID-19 because they induce immunity against the SARS-CoV-2 virus, that is, they reduce the risks of contagion, becoming symptomatic, developing severe forms of the disease, and dying [12]. At this point, it should be recognized that not all countries have the necessary personnel and vaccines; this is especially true of less developed countries [13].

After 26 months since the start of the pandemic and 16 months of administering vaccinations, despite the progress that has been made, we cannot say that we are close to overcoming the COVID-19 crisis; the danger persists since there is inequality in the distribution of vaccines, despite the efforts made by governments. However, we cannot lower our guards, and although some countries have high vaccination rates, they are not invulnerable. The emergence of new variants that are resistant to vaccines is a possibility and could represent a setback in the fight against SARS-CoV-2 [14].

It must be recognized that this problem is related to inequality in socioeconomic determinants, such as the gross domestic product of each country, considering that countries with fewer resources receive fewer vaccines [15]. Therefore, we must remember that as long as there are countries with low vaccination rates, the risk of virus shock and transmission to their populations increases, a problem that transcends every country’s border [16]. It is for this reason that efforts must be increased to reduce this inequality and increase the speed of vaccination worldwide. Governments must propose coordinated policies that are enforced by international authorities [17].

Chakraborty et al. [18] identified a reluctance to get vaccinated in Asia, where, despite a vaccine efficacy rate between 70% and 95%, the population expressed low acceptance, with the highest acceptance being related to well-organized campaigns. Cesaroni et al. [19] found that only 10.3% of the subjects in their study were not vaccinated, and this result was associated with having a chronic disease, being a foreigner, and having a low education level. Not being vaccinated was related to receiving unclear information and having fears due to misinformation; as such, information about vaccination should be presented differently according to the social group to which it is directed. On the other hand, in a qualitative study on vaccine acceptance among black people from the United Kingdom and the United States, Ogueji et al. [20] found cultural obstacles; a mistrust of vaccines and the speed with which they were produced; poor policies, especially regarding vaccine delivery; and historical factors such as a history of discrimination and abuse of black people in the medical community to be influencing factors.

Sua et al. [21] identified the effects of socioeconomic factors on health (which have been recognized by the WHO), and hence, their value. The Human Development Index (IDH), GINI, longevity, gross domestic product (GDP), and other factors have influenced the effects of this pandemic. For example, there is an association of mortality rate and infection due to COVID-19 with the GINI [21]. To control the pandemic, it is necessary that countries with low vaccination rates improve their vaccination coverage, taking into account existing inequalities and bearing in mind that countries with the highest HDIs have a better coverage. In addition, it is necessary that international collaboration takes place between all countries and vaccine producers, and that we develop effective communications systems [13].

In this research, socioeconomic, biological (age over 65 years, average life expectancy, and average age in years), vaccination-related (one dose, two doses, and three doses), and pandemic-related (cases and deaths) variables were examined. An analysis using quartiles was carried out, and the Concentration Index was calculated for each dose and socioeconomic indicator. Countries were grouped by continent, and we analyzed vaccination doses using a Lorenz curve; then, vaccination rate of each country was correlated with the Human Development Index (HDI) and the Fragility Index of Nations (IFN). The temporal aspect was also considered when assessing the accumulated percentage of vaccinated people by region and by continent. The Competitiveness Index of Nations (ICN) and the Fragility Index of Nations measure the behavior of a state regarding its decisions; it was not possible to find studies that use these indicators when assessing state decision making. Researchers have proposed the use of decision-making tools by authorities, such as the Concentration Curve and Index, the Lorenz curve, and the GINI Index, which would serve to measure the effects of the present pandemic and manage future pandemics. These tools are oriented toward assessing decision making and the elimination of inequalities and inequities in the shortest possible time, with the transfer of goods to the less fortunate, considering that all goods can be transferred [22]. This study aimed to determine the correlations between certain COVID-19-related socioeconomic determinants, cases, and deaths with vaccination worldwide, and to measure inequalities in vaccination over time and across geographic regions.

## 2. Materials and Methods

### 2.1. Ambit

In the database used, Our World in Data, only 224 countries are considered, of which those with more than 500,000 inhabitants were selected; countries with fewer than 500,000 inhabitants are generally very small or include islands that do not have data on the socioeconomic indicators included in this research. Ultimately, 169 countries were chosen (52 from Africa, 26 from North American and South America, 39 from Europe, and 52 from Asia and Oceania) for this study. This decrease in the number of countries did not impact external validity because the excluded countries’ populations are small. Oceania is associated with Asia because it includes few countries. For the temporal analysis of vaccination, the groups of countries specified in the database were used. Regarding time, this study considered its starting point to be the time the pandemic was declared on 30 January 2020 and its end point to be 31 March 2022. Vaccination was measured from the date of its initiation in each country and each group of countries.

### 2.2. Design

A study of secondary sources was carried out to assess the inequality in vaccination against COVID-19 worldwide in association with socioeconomic factors. A general analysis of the socioeconomic indicators was conducted, which consisted of a descriptive analysis of the variables, and the normality and collinearity of the variables were investigated.

The quartiles, medians, and means were calculated. Next, correlations between SARS-CoV-2 vaccination and socioeconomic indicators, cases, and deaths were determined, and then, a table was prepared to analyze vaccination by continent. All these procedures served to enable us to choose the relevant mathematical models. A panel of the Lorenz curve (GINI Index) was developed to analyze vaccination with the 1st, 2nd, and 3rd doses on the continents of Africa, North and South America, Europe, and Asia/Oceania.

Concentration Curves were created by ordering the vaccination percentage variables using the aforementioned socioeconomic indicators. In the same way, a table was prepared that showed us the Concentration Index by continent and the socioeconomic variables’ competitiveness, HDI, and fragility with respect to vaccination.

Regarding the units of analysis, the continents of Africa, North and South America, Europe, and Asia/Oceania were considered, as was the age group of the elderly, as defined above. Population and sample: populations were grouped by their country of belonging and were further grouped into those who had contracted COVID-19, those who had died, and those who had been vaccinated with the 1st, 2nd, and 3rd doses; countries were also grouped by continent (Figure 1).

### 2.3. Variables

Vaccination with 1, 2, and 3 doses: (1) vaccinated with one dose: people vaccinated with one dose per 100 people in the total population; (2) vaccinated with two doses: people vaccinated with two doses per 100 people in the total population; and (3) vaccinated with three doses: people vaccinated with three doses per 100 people in the total population. This study did not consider the types of vaccines received around the world [23].

Global Competitiveness Index 2019 (ICG): This index comprises the attributes and qualities of an economy that allow for the most efficient use of production factors. It measures how a country uses its resources and its ability to provide its inhabitants with a certain level of prosperity [24].

Human Development Index 2019 (HDI): This is an indicator used to measure the progress of a country that analyzes health (life expectancy at birth), education (years of schooling), and wealth (dignified life). The Human Development Index HDI 2019 [25] was used.

Fragility Index of Nations 2021 (IFN): this index evaluates the vulnerability of countries using twelve indicators (cohesion, economic, political, and social) to measure the condition of a state at a given moment [26].

Corruption Appreciation Index (IAC): this is the percentage of people who consider that the use of public power for private benefit is improper [27].

Gross Domestic Product per person in 2020 (GDP).

Doctors per thousand inhabitants (World Bank): the number of doctors in a country per one thousand inhabitants [28].

Aged 65 and over: the percentage of a country’s population that is aged 65 years or over, using the most recent data available [23]. 

Life expectancy at birth: the median age of a population in years (UN projection for 2020) [23].

Cases: confirmed cases of COVID-19 per 100,000 inhabitants, considering any test available in the country of reference, including probable cases [23].

Deaths due to COVID-19: deaths due to COVID-19 per 100,000 inhabitants, considering any test available in the country of reference, including probable cases [23].

Continents: the geographic location in terms of continent of the included countries: 52 from Africa, 26 from North and South America, 39 from Europe, and 52 from Asia and Oceania.

Continents for temporal analysis: this analysis included 51 countries from Asia, 51 countries from Europe, 37 countries from North America, 13 countries from South America, 23 countries from Oceania, and 56 countries from Africa.

Health inequalities measure quantifiable differences in a certain dimension in groups of individuals, including inequity, which is unfair inequality that can be prevented, modified, or avoided [29,30].

### 2.4. Analysis

For the exploratory analysis, the following were used: LibreOffice, JASP 0.16.1.10., and mathematical models, such as normality analysis (with Shapiro–Wilk test, QQ curve, bias, and kurtosis). The data were evaluated, and it was verified that they were complete; when some data were found to be missing, they were adjusted according to their regions, and we calculated the averages of the countries by region only if the deficiency was less than 10%. The univariate analysis was performed using the programs LibreOffice and JASP 0.16.1.0, through which we calculated the quartiles, means, extreme values, frequencies, proportions of those vaccinated, and quartiles of mortality and lethality. For the bivariate analysis, the following non-parametric bivariate statistical tests and mathematical models were used: a Kruskal–Wallis test, Spearman’s correlation, a temporal analysis of vaccination, a Lorenz curve, a Concentration Index, and a slope. For this, the following programs were used: EPIDAT 4.2, LibreOffice 7.2., and JASP 0.16.1.0. Correlations between SARS-CoV2 vaccination and the socioeconomic indicators, cases, and deaths were determined; then, a table was prepared to analyze vaccination by continent. A panel of the Lorenz curve (GINI Index) was developed to analyze vaccination with 1, 2, and 3 doses on each continent (Africa, North and South America, Europe, and Asia/Oceania), and we examined for possible inequalities across countries. A multivariate analysis was carried out using the Concentration Curve, through which we crossed the variables of vaccination percentage, population, and socioeconomic indicators. In the same way, a table was created that shows the Concentration Index by continent and by socioeconomic variable. The units of analysis were the countries and their characteristics, as well as the continents of Africa, North and South America, Europe, and Asia/Oceania.

Ethical aspects: Since the data were obtained from secondary sources of international organizations, no request for permission was required. Given that this is an ecological study, it was not required to go through an ethics committee. There was also no contact with subjects, so informed consent was not required.

## 3. Results

A total of 169 countries were considered in this study (52 from Africa, 26 from North and South America, 52 from Asia/Oceania, and 39 from Europe). Differences were found between the mean and the median of all the variables (the data did not have a Gaussian distribution); the median of having three doses of vaccine was 10.22% and the mean was 20.72%; other variables had greater differences, such as GDP 2020. A difference was found between the 25th and 75th quartiles (17 points) of the 2019 ICG, with great dispersion between the countries; the same was demonstrated upon analyzing the behavior of the other indicators.

The countries located in the fourth quartile (75th percentile) had higher values for positive indicators, such as the GCI, HDI, number of physicians per 1000 inhabitants, GDP, and life expectancy, compared to those located in the first quartile. For some indicators, there was a large difference between the values of the countries in the 25th and 75th percentile; for example, the value for doctors per one thousand inhabitants was 3.103/0.375, which means that 25% of the countries have less than one doctor per 1000 inhabitants, with a difference of 8.27 times more doctors per 1000 inhabitants in more developed countries; the GDP (15,720.99 vs. 1701.99) showed a nine-fold difference, indicating that in some countries, people earn less than USD 1701.99 per year, and in others, they earn nine times more. Other indicators showed smaller differences between countries, such as the ICG 2019 (66.20 vs. 49.30). IFN 2021 measures to what extent a nation is vulnerable to having a higher Fragility Index, which means that those countries above the 75th percentile are more fragile; in our case, the 75th percentile reached 84.10 points. Regarding average age, 25% of the countries had a young population with an average age of less than 21.50 years, showing a difference of 17 years between the countries in the 25th and 75th percentiles (21.50 vs. 39.10 years). In relation to vaccination, there is a need for higher rates of vaccination with one, two, and three doses. The differences between countries in the 25th and 75th percentiles were also marked; the higher the number of doses, the lower the value of the 25th and 75th percentiles. The difference for one dose was (77.19/26.92, reason 2.87); for two doses, it was (72.11/18.72, reason 3.85); and for three doses, it was (38.22/0.36, reason 106.17), which indicates that the higher the number of doses, the greater the difference between countries. When analyzing the percentage of vaccinated people in the countries in the 25th percentile, we found that the number of people who had been vaccinated with a first dose was 75 times more than the number of people who had been vaccinated with a third dose; in countries in the 75th percentile, the number of people who had been vaccinated with a first dose was 2.01 times more than the number of people who had been vaccinated with a third dose; these results indicate a great difference between the countries in the 25th and 75th percentiles. When comparing the medians, it can be seen that the higher the number of doses, the lower the percentage of vaccinated people (*p* < 0.05) (Table 1).

Significant correlations were found between vaccination with one, two, and three doses and all the socioeconomic variables, the values of which increased as the number of doses increased. The correlation was positive with the ICG, IDH, IAC, number of physicians per 1000 inhabitants, GDP, age over 65 years, life expectancy, and average age in years, and it was negative with the IFN. Life expectancy had the highest correlation with vaccination with three doses and increased with increasing number doses, which shows that in countries with a higher life expectancy, there is a higher rate of vaccination. For the HDI 2019, we can say the same, as with the number of physicians per 1000 inhabitants, GDP 2020, age over 65 years, GCI, and average age in years. The IAC (Corruption Appreciation Index) result is interesting and shows that the greater the perception of corruption, the more people were vaccinated, and this increased with the number of doses of vaccine. With the Concentration Index, which measures inequities in vaccination, it was noted that for all doses, there were inequalities between countries, and these were related to the socioeconomic indicators; it was also observed that greater inequality existed for receiving a higher number of vaccine doses (Table 2).

In the groupings of countries by continent, the results of the GINI Index also showed inequality in vaccination against COVID-19 with one, two, and three doses in all continents, with greater inequality as the vaccine dose increased (GINI associated with receiving third dose of vaccine: Africa: 0.815, Asia/Oceania: 0.500, Europe: 0.332, and North and South America: 0.248). There were differences between the countries in each continent, with Africa being the continent with the highest degree of inequality (GINI associated with receiving one dose: 0.430, two doses: 0.476, and three doses: 0.815); this could mean that in African countries, in addition to having few vaccines, the availability of vaccines is concentrated in certain countries, indicating an unequal distribution of vaccination. There was less inequality in North and South America, which had the following GINI indices: first dose: 0.074, second dose: 0.089, and third dose: 0.248. Vaccination inequalities existed in Europe, with the GINI indices being higher than those in the Americas and Asia/Oceania (0.128, 0.134, and 0.332 for one, two, and three doses in Europe). The continent with the highest inequality in vaccination was Africa, followed by Europe, Asia/Oceania, and lastly, the Americas.

The Lorenz curve showed inequality in the distribution of vaccines for one, two, and three doses on all continents, with inequality being lowest for the first dose, increasing for the second dose, and greatest for the third dose. With the first dose, North and South America showed a lower inequality, followed by Asia/Oceania and Europe, and Africa had the greatest inequality; this result was repeated with the second dose. For the third dose, the increase in inequality was greater in Africa and Asia/Oceania. Comparing the increase in inequality for the third dose with that for the second dose, it was found that inequality is greater in Asia/Oceania than in Africa, but the slope of the segments of the Lorenz curves for doses is more pronounced in Africa, since 70% of the population in Africa received only 10% of the vaccines available in the continent. In Europe, 40% of the population received only 10% of the third doses of vaccines available, and in Asia/Oceania, 55% of the population received only 10% of the vaccines available (Figure 2).

The Vaccination Concentration Index was obtained from the Concentration Curve by ordering the vaccination results of the countries of each continent according to the indicators of competitiveness, HDI, and Fragility Index of Nations (this allowed us to analyze the correlation between the ordering of countries according to socioeconomic indicators and vaccination, and to interpret whether inequality is related to these indicators). The results showed that vaccination with one, two, and three doses was more concentrated in continents that had the best socioeconomic indicators, with greater inequality occurring as the number of vaccination doses increased. The Vaccination Concentration Index with one dose ordered by competitiveness was higher in Africa (0.175) than in Europe (0.100) and Asia/Oceania (0.070), and lower in America (0.010). With two doses, this pattern remained the same for each continent. With three doses, a higher concentration was found for Africa, followed by Asia/Oceania (0.387) and Europe (0.285), and a lower concentration was found for America (0.035); this means that vaccination is more concentrated in some countries on each continent, with this inequality being greater in Africa and being related to the Competitiveness Index of Nations. The same happened with vaccination ordered according to the IDH, for which a higher concentration was found in the countries with the highest Human Development Indices. The same happened with vaccination ordered according to the Fragility Index, with this concentration being higher in some African countries. On all continents, the highest concentration occurred for vaccination with the third dose (greater inequality). When comparing North and South America with Europe, it is interesting that of the three socioeconomic variables, the concentration indicator was lower in North and South America than in Europe, meaning that there were higher concentrations in some European countries, which occurred less than in North and South America (Table 3).

The results of the analysis of vaccination quartiles by continent were similar. There was no equality in the distribution of vaccination, and the median was greater for North and South America, followed by Asia/Oceania, Europe, and, then, Africa, with the greatest dispersion in Asia/Oceania.

Vaccination began in November 2020, with Asia, North America, and Europe being the first to vaccinate their populations, and Africa being the last. In the figure, all continents show a temporary upward vaccination curve, with their inclinations or slopes of vaccination varying; thus, Africa has a lower vaccination rate than the other continents (which had administered vaccinations more quickly). This pattern is repeated for vaccination with one, two, and three doses, reflecting that the percentages of the vaccinated population are lower in Africa (21%, 12%, and 2.5% for each dose). There is a great difference between the vaccination slopes of Africa and Oceania, which had vaccinated people up to six times faster (Figure 3).

The rate of vaccination with the first dose was slow in the first months of the pandemic on all continents, but then, the slope improves, increasing in South America (6.67%/month) and in Oceania (6.07%/month), unlike Africa, where the vaccination rate was 1.52%/month. South America was the region in which the highest percentage of the population was vaccinated (82%), followed by Asia (72%), North America (70%), Europe (68%), and Oceania (66%), while Africa only managed to vaccinate 20% of its population. Regarding the second dose, South America has the best slope (6.19%/month, 4.3 times faster than Africa), vaccinating 72% of its population; the other continents vaccinated between 62% and 68% of their populations, unlike Africa, which vaccinated only 16%. Regarding the third dose, Oceania had the best vaccination rate of 8.28%/month, followed by Europe (4.49%/month), and Africa (0.22%/month); Europe was the continent that vaccinated the largest percentage of its population at 38%, followed by Oceania at 37%, South America at 34%, North America at 25%, Asia at 25%, and finally, Africa, which vaccinated only 2% of its population; all of these data were collected up to March 2022 (Figure 3).

## 4. Discussion

### 4.1. Important Findings

In this investigation, a large difference was found between countries in the first and fourth quartiles in all the indicators studied, which indicated that for vaccination at a higher dose, there was a greater difference in the number of people vaccinated with one, two, and three doses. The correlation was significant between vaccination with one, two, and three doses and all the variables. The greater the number of doses, the greater the correlation. When measuring the Concentration Index for each variable, we found that they showed the same behavior as the correlations. The Corruption Index result is interesting; the higher the Index in a country, the more people in that country were vaccinated. Upon assessing the Lorenz Curve, we found greater inequality in Africa compared to the other continents. On all continents, greater inequity was found with regard to the third dose than with regard to the second and first doses of vaccination. Regarding the Concentration Index, the higher the number of doses, the more this Index increased, which indicated inequality in vaccination on the continents. In Africa, there was a greater concentration, and in America, there was a lower concentration. The three socioeconomic indicators (competitiveness, the HDI, and the Fragility Index) showed similarity, leading to the conclusion that there is a presence of inequality in vaccination between the continents. Vaccination began one year after the start of the pandemic. The last continent to start administering vaccinations was Africa, which presented a very flat slope (very slow vaccination). The continent that vaccinated the most people was South America.

### 4.2. Comparison with Other Studies

Socioeconomic indicators that are used as determinants of health, since they are modifiable and related to vaccination against SARS-CoV-2, should be taken into consideration to establish plans to improve vaccination coverage and reduce inequalities; this emphasizes the importance of their identification. In this regard, Sua Dy et al. [21] studied socioeconomic and demographic variables that were similar to ours and assessed their relationships with the general health of populations; they found that countries that suffered the highest burden of disease from COVID-19 had a more unequal income distribution. Other investigations [31,32,33] have also found an association between vaccination and socioeconomic determinants, which favor populations with better indicators; such results were obtained in the study by Basak et al. [32], who found an association between GDP and vaccination, with better rates in higher-income countries. In North America [34], inequality in vaccination among African Americans was found to occur due to factors such as race, ethnic origin (first doses were administered in 60% of Whites and 9% of African Americans). Duan et al. [15] identified that vaccination coverage varied by income group, with vaccinations being administered 2.33 times more people with high incomes. Basak et al. [32] found that the richer a country, the higher the vaccination rate; in countries that earned more than USD 22,000 per year, vaccination coverage was higher than 89%. Brown et al. [35] evaluated the difference in vaccination between those over 18 years of age and those over 65 years of age, finding that there were more vaccinated people over 65 years of age; however, this could have been because those over 65 years of age were a population prioritized for vaccination. The quintiles of the socioeconomic variables did not differ much. Other studies, such as that of Brown et al. [35], found an association between vulnerability and vaccination rates, which were higher in countries with less vulnerability; this was also reported by Bruckhaus et al. [36], showing agreement with our research. On the other hand, Spreco et al. [37] found an association between the Corruption Perception Index and vaccination coverage; a 1% increase in the perception of corruption was associated with increased vaccination by 0.45% as long as the HDI remained stable with a positive correlation, as found in our work; this is interesting and may be due to the fact that at all stages of vaccination and at all levels (from the start of manufacturing to acquisition, dissemination, and distribution), there are possibilities of corruption, which would jeopardize countries’ goals and lead to fewer resources for achieving rapid vaccination [38].

Another investigation found an association between the HDI and vaccination coverage, with better results obtained when continents had a higher HDI, as in the case of Europe compared to Africa [13]. Roghani [33] found no association of vaccination with the HDI but an association with life expectancy, as in our research. It was also found that the higher the number of physicians, the more people were vaccinated [39]. Sina-Odunsi [34] studied vaccination inequities while considering ethnic groups, finding African Americans, Latinos, and Asians being the least advantaged. Bruckhaus et al. [36] investigated the relationship between people’s vulnerability and vaccination, and found a greater disparity in vaccination coverage between the categories of low and high vulnerability. Williams et al. [31] studied disparities in vaccination while considering White, Black, and Asian populations, finding that the most favored were Whites and Asians; this was also associated with the acceptance of the vaccine. Brown et al. [35] found that increased vulnerability of a community was associated with low vaccination rates.

In our case, the percentage of people vaccinated with one, two, and three doses had a positive correlation with the socioeconomic indicators ICG, IDH, IAC 2021, physicians per 10,000 inhabitants, GDP 2020, age over 65 years of age, average life expectancy, and negative consequences with the IFN. Here, a positive association with the Corruption Appreciation Index was confirmed. On the other hand, the Concentration Index revealed that the countries that had the lowest levels for the socioeconomic indicators had a lower proportion of vaccinated people in their population. Regarding the third dose, 50% of the population with the lowest HDI comprised only 9% of those vaccinated, and 50% of the population with the lowest GDP per capita comprised 10% of those vaccinated with three doses. Other investigations also found positive correlations; for example, Moitre et al. [2] found a correlation between IDH and vaccination; Duan et al. [15] found a correlation with a wealth indicator, Basak et al. [32] found a correlation of around 0.98 with socioeconomic variables, and Roghani [33] found correlations with poverty, life expectancy, and HDI.

In our geographical analysis, it was found that inequality in vaccination against SARS-CoV-2 was greater with regard to the third dose in Africa, where 50% of the countries had vaccinated less than 10.22% of their populations within the timeframe of this investigation. The Lorenz curve also shows inequality in vaccination with one, two, and three doses. This inequality, similarly, was greater with regard to the third dose on all continents; Africa was the most affected, unlike America, which had a more equal distribution. The GINI Index showed similar results to the Lorenz curve (GINI Indices associated with the second dose for Africa, Europe, Asia/Oceania, and America were 0.476, 0.134, 0.129, and 0.089, respectively). No previous research was found on this topic that used these tools to measure inequalities.

Regarding the third dose, the Vaccination Concentration Index, in the order of the ICG, IDH, and IFN, showed that vaccination inequality was concentrated in countries on the most socioeconomically advantaged continents, with the greatest inequality occurring in Africa. South America had the highest proportion of vaccinated people, followed by Asia, Europe, Oceania, North America, and Africa. Regarding three doses of vaccination, the continent with the highest proportion of vaccinated people was Europe, followed by Oceania and South America. Researchers such as Basak et al. [32], using other tools, identified that Africa was the country that was furthest behind in terms of vaccination, and that wealthier nations had higher vaccination rates regardless of their continent; thus, Eastern European countries had lower vaccination rates.

Other investigations, such as that of Ning et al. [13], also found higher vaccination rates in developed countries compared to Africa. Rhogani [33] carried out an investigation over two periods of time; in the first period, he did not find a relationship with the IDH, and in the second period, a 31% increase in vaccination was found.

### 4.3. Actions That Could Be Taken

It is important that countries quickly vaccinate more than 80% of their populations to allow their economies to reactivate; on the other hand, in countries that have good economic stability, the problem is not the availability of vaccines but their acceptance. The WHO should address how to improve the equitable distribution of vaccines, aiming to increase their availability in countries that have vaccinated the fewest people, especially at-risk groups; this could lead to the reactivation of these countries’ economies [40]. Omicron is the variant that circulates throughout the greatest parts of the world [41]; fortunately, two doses of vaccine are still effective at protecting populations; however, this protection is not permanent and decreases as time goes by, which is not the case for all available vaccines.

For Omicron, as the protection provided by one dose of vaccine is not complete, it is recommended to give three doses to immunosuppressed and elderly people to protect them against circulating variants. Despite these findings and the efforts made by some countries, countries are unable to achieve high vaccination rates with two doses and achieve even lower rates with three doses; this should be a matter of concern since only vaccination with three doses can provide an adequate level of protection [42]. Faced with the inequitable distribution of vaccines, an alternative solution could be to require vaccine manufacturers to share their technology and knowledge and, thus, enable vaccines to be manufactured in low-income countries, which would allow better and more equitable distribution in these regions [43]. Africa, which is the continent that has vaccinated the fewest people, produces only 1% of its vaccines and has reached agreements to increase their production; however, countries in this region prefer to use only donor vaccines [44]. On the other hand, some more developed countries support policies that promote equality in distribution of vaccines; these countries state that donations to underdeveloped countries that are unable to vaccinate their people should be prioritized before providing booster vaccinations to developed countries, as promoted by the WHO, to achieve control of COVID-19 [45]. Often, the recommendations of international organizations regarding the principle of equality are not followed, which dictates that countries must comply with policies guaranteeing universal access and equity in distribution of vaccines to countries that cannot meet their needs; hence, access continues to be insufficient in low- and middle-income countries compared to high-income countries in relation to both the quantity of vaccines and their price, as observed in Latin America [46]. Vaccination with two doses provides good protection, especially when two types of vaccine are administered, but a third dose is required, and sometimes a fourth, to give greater protection against the Omicron variant, although it is not known how long this protection lasts [47]. We must remember that a vaccination schedule against this disease is still being built. Researchers such as Atuesta-Montes et al. [22] point out that there are tools to measure inequality and plan redistribution included in social policies. Duan et al. [15] found that the stronger a country’s vaccination policy, the higher its vaccination coverage, and the opposite occurs in low-income countries with weak policies that are more limited in their vaccination capabilities.

This research highlights the problems of inequality in vaccination worldwide, despite the actions taken by leaders and decision-making authorities, as well as the important role played by socioeconomic indicators. It is necessary for governments to consider the use of the Lorenz and Concentration curves, which would allow inequality in vaccination to be assessed in each country and region, and for groups to prioritize implementing policies and interventions based on their results, with the aim of improving vaccination coverage and reducing existing inequalities. This is not a problem that can be solved at the individual level but, rather, at the government level so that everyone has the same opportunity to be vaccinated, which would reduce, among other things, the risk of the appearance of new strains.

Limitations: Regarding the limitations of this research, we must consider that the data were obtained from secondary sources of different origins and evaluated using instruments designed for other purposes. The pandemic evolved quickly, as did vaccination. The variables taken into account, due to the characteristics of their non-Gaussian distribution, did not allow the use of parametric statistical tests and mathematical models. There was also no participation in the preparation of the instruments or in data collection; the definitions were given by international organizations and had to be taken as they were. One of the limitations of this type of study whereby data are obtained from secondary sources is the risk of registration errors in data temporality and completeness; this has happened more in some countries than in others. Another disadvantage is the inability to identify whether the influencing variables precede the outcome variable, so it is necessary to interpret the results in context. Moreover, the reduction in information does not allow individual analysis. Multicollinearity makes it difficult in multivariate analyses to identify the most significant variables.

## 5. Conclusions

There was inequality in vaccination with one, two, and three doses against COVID-19, and this was associated with socioeconomic indicators; thus, the countries with the highest number of cases and deaths had unfavorable indicators, as well an insufficient level of vaccination. The ICG had a positive correlation with vaccination, unlike the IFN, which had a negative correlation. Inequality, measured using the Lorenz curve, was higher in Africa, followed by Europe, Asia/Oceania, and finally, North and South America, increasing as number of vaccine dose increased; regarding the third dose, its uptake was very low, with the lowest value occurring in Africa. The rate of vaccination was slow in the first months, with South America being the region that vaccinated the most people with the first dose, followed by Asia, North America, Europe, Oceania, and Africa. This region vaccinated a lower percentage of its population with the second and third doses.

### Recommendations

We recommend continuing to use socioeconomic variables to measure inequalities in vaccination that may exist in each country. We also recommend improving health systems, preventing increases in existing inequality in vaccination, and using policies and strategies based on information obtained using appropriate indicators and tools.

## Figures and Tables

**Figure 1 vaccines-11-01245-f001:**
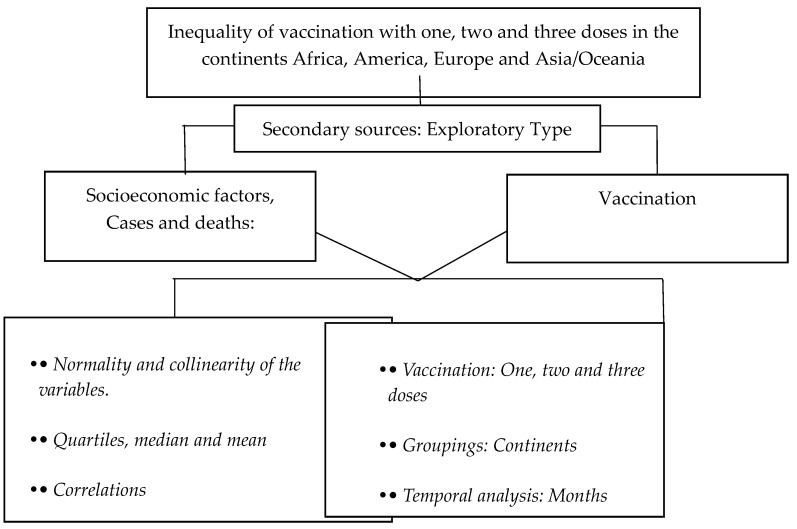
Research design flowchart.

**Figure 2 vaccines-11-01245-f002:**
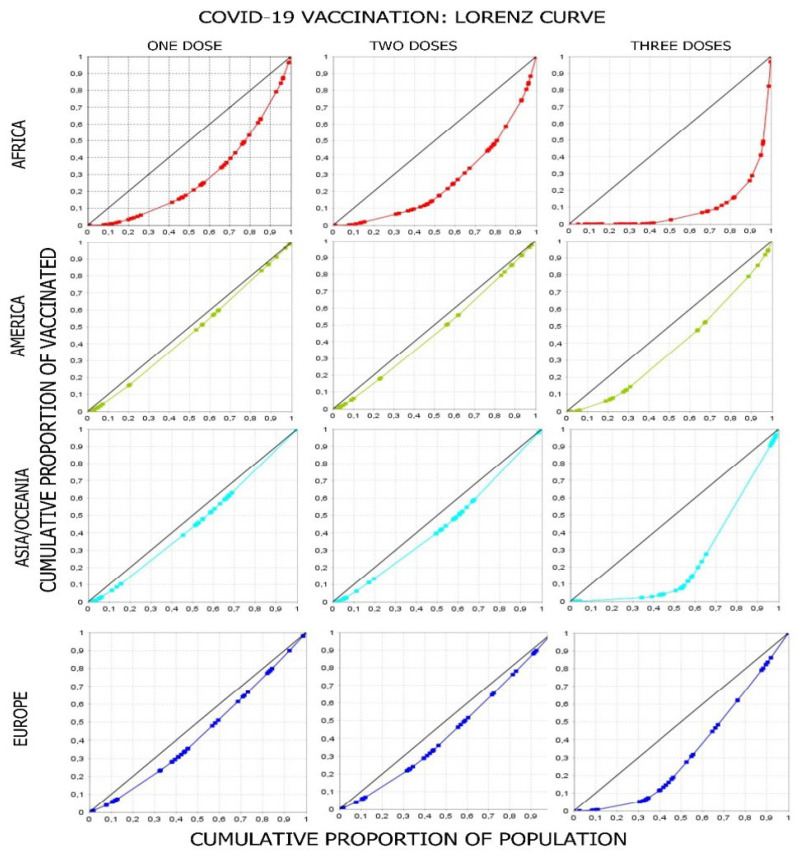
Lorenz curve of COVID-19 vaccination worldwide.

**Figure 3 vaccines-11-01245-f003:**
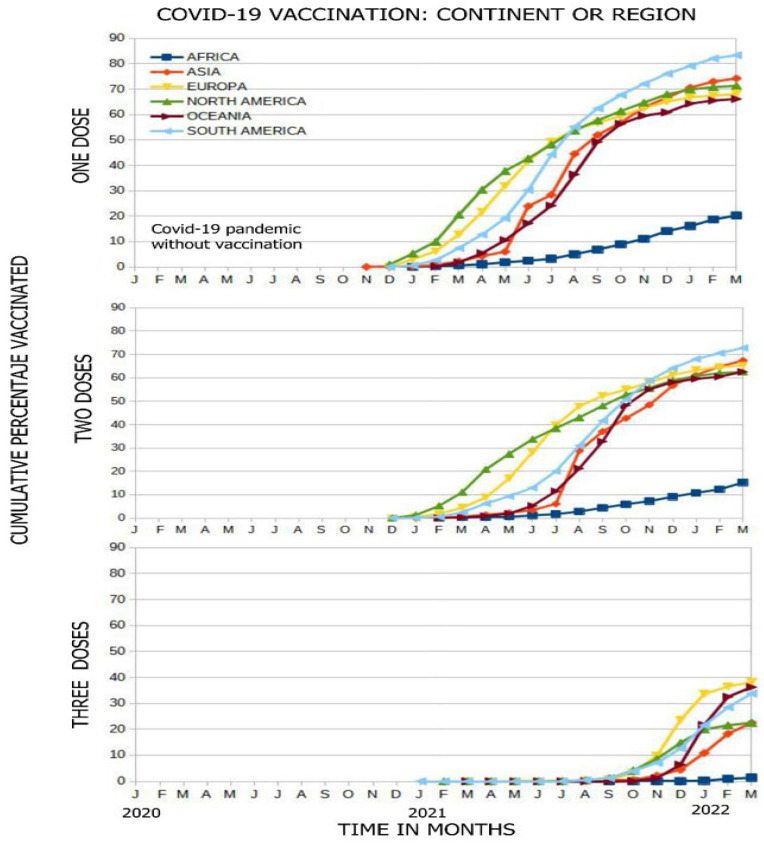
COVID-19 vaccination on each continent according to dose per month, from the start of vaccination.

**Table 1 vaccines-11-01245-t001:** Quartiles of COVID-19-related socioeconomic indicators, vaccinations, cases, and deaths worldwide.

Variables	Quartiles			
	P25	P50	P75	Median
ICG 2019	49.30	57.50	66.20	58.49
IDH 2019	0.58	0.74	0.81	0.72
IFN 2021	51.00	70.00	84.10	67.36
IAC 2021	29.00	39.00	54.00	42.65
Doctors per 1000 people	0.375	1.719	3.103	1.95
PIB 2020	1701.99	4603.34	15,720.99	13,383.84
Age over 65 years, %	3.38	6.21	14.36	8.77
Life expectancy in years	66.60	74.08	78.49	72.48
Average age in years	21.50	29.30	39.10	30.32
One dose, %	26.92	59.74 *	77.19	52.55
Two doses, %	18.72	50.72 *	72.11	47.56
Three doses, %	0.36	10.22 *	38.22	20.72
Cases per 100,000 people	677.16	7209.92	16,514.14	11,641.65
Deaths per 100,000 people	11.56	65.39	180.76	114.12

GCI 2019 = Global Competitiveness Index 2019; HDI 2019 = Human Development Index 2019; IFN 2021 = Fragility Index of Nations; IAC = Corruption Appreciation Index; GDP 2020 = Gross Domestic Product per Person; One dose = % of population vaccinated with one dose; Two doses = % of population vaccinated with two doses; Three doses = % of population vaccinated with three doses; * Kruskal–Wallis, *p* < 0.05.

**Table 2 vaccines-11-01245-t002:** Correlations of vaccination against SARS CoV2 with socioeconomic indicators, cases, and deaths.

Variables	Spearman’s Correlation			Concentration Index		
	One Dose	Two Doses	Three Doses	One Dose	Two Doses	Three Doses
ICG 2019	0.68 **	0.72 **	0.76 **	0.16	0.19	0.41
IDH 2019	0.74 **	0.79 **	0.84 **	0.15	0.17	0.44
IFN 2021	−0.69 **	−0.75 **	−0.79 **	0.15	0.17	0.43
IAC 2021	0.64 **	0.70 **	0.73 **	−0.14	−0.17	−0.39
Doctors per 10,000 people	0.64 **	0.69 **	0.73 **	0.14	0.16	0.42
PIB 2020	0.71 **	0.76 **	0.82 **	0.14	0.17	0.48
Age over 65 years, %	0.59 *	0.63 **	0.67 **	0.16	0.19	0.44
Life expectancy in years	0.77 **	0.81 **	0.85 **	0.16	0.19	0.50
Average age in years	0.69 **	0.74 **	0.77 **	0.16	0.20	0.47
Cases per 100,000 people	0.61 **	0.66 **	0.75 **			
Deaths per 100,000 people	0.41 **	0.45 **	0.53 **			

** = *p* < 0.0001; * = *p* < 0.001.

**Table 3 vaccines-11-01245-t003:** Concentration Index of vaccination on the continents according to socioeconomic indicators.

Continent/Vaccination	Socioeconomic Variable Concentration Index		
	Competitiveness	IDH	Fragility
**AFRICA**			
One dose	0.175	0.252	0.224
Two doses	0.202	0.257	0.262
Three doses	0.542	0.566	0.570
**AMERICA**			
One dose	0.010	0.022	0.017
Two doses	0.012	0.029	0.027
Three doses	0.035	0.027	0.107
**ASIA/OCEANIA**			
One dose	0.070	0.065	0.071
Two doses	0.098	0.093	0.093
Three doses	0.387	0.442	0.355
**EUROPE**			
One dose	0.100	0.102	0.085
Two doses	0.106	0.109	0.095
Three doses	0.285	0.294	0.270

## Data Availability

The data used in this study are available from the authors upon reasonable request.

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
