# Peer review of "Global Inequities in COVID-19 Vaccination: Associated Factors and Tools to Measure Inequality"

_vaccines, 2023, doi:10.3390/vaccines11071245_

Round 1

Reviewer 1 Report

The topic for investigation is both timely and important for present reflection on meeting the challenges of COVID-19 and thereby providing some insight as to future dynamics with other infectious diseases worldwide. The study is highly salient in this regard. The Introduction provides a sufficient rationale but is short in making a compelling argument. Some of the references in the Discussion could more appropriately be included in the Introduction. Ideally, the authors should include more specific socioeconomic concepts and categories in the Introduction to justify the use as variables for analysis. For example, population characteristics and quite different from the Fragility Index of Nation or “continents of geographic location.”  This has, among other things, implications for conclusions and further actions.

The study makes effective use of multiple data sources as evidenced by the figures and the analysis tables. The analysis, statistics and presentation are appropriate.

The analysis is based on 169 countries. I noticed that data was available for 224 countries. Some brief explanation for excluding countries is in order. Further, does this feature impact on the external validity of the results?

“Inequality” and “inequity” terms are used. Yet I do not find a clear definition and operationalization of the two terms and how they are applied in both the Results and Discussion sections.

The Discussion, while interesting, could be improved with more judicious editing and summarization so that it leads more directly to the Conclusions. It is difficult to align the Conclusions with the Discussion section.

Limitations of the study are not mentioned in terms of internal and statistical conclusion validity nor is external validity adequately discussed.

Author Response

Dear Reviewer, 1,

We update the article according to your indications.

The changes suggested by the reviewers there are in color red.

Best regards,

The authors

Reviewer 2 Report

This is a good work informing actions that must be taken to reduce COVID-19 vaccination inequalities. I only suggest you connect your study and findings with a related work: 

Ogueji, I. A., Demoko Ceccaldi, B. M., Okoloba, M. M., Maloba, M., Adejumo, A. O., & Ogunsola, O. O. (2022). Black People Narrate Inequalities in Healthcare Systems that Hinder COVID-19 Vaccination: Evidence from the USA and the UK. Journal of African American Studies26(3), 297-313. https://doi.org/10.1007/s12111-022-09591-5

Author Response

(The authors gave the same response as above.)

Reviewer 3 Report

The paper aimed to correlate some socioeconomic determinants, cases, and deaths from COVID-19 with vaccination in the world, as well as measure inequalities in vaccination over time and geographic regions. It can be accepted for the publication after some major revisions.

1. Justify the choice of materials and methods used in your study.

2.  Check the language and punctuation of the manuscript.

3. Compare your results with others existing in the literature.

4. Give more interpertations of the tables and figures presented in your paper.

5. The conclusion section should be rephrased and written clearly.

Check the language and punctuation of the manuscript.

Author Response

Dear Reviewer 3,

We update the article according to your indications.

The changes suggested by the reviewers there are in color red.

Best regards,

The authors

Round 2

Reviewer 1 Report

With pleasure I read the revisions made by the authors.  These revisions substantially improved the analysis, results and discussions.

Reviewer 3 Report

The paper can be accepted.

The paper can be accepted.